# Barriers to Seeking Mental Help and Interventions to Remove Them in Medical School during the COVID-19 Pandemic: Perspectives of Students

**DOI:** 10.3390/ijerph19137662

**Published:** 2022-06-23

**Authors:** Barnabás Oláh, Bence Márk Rádi, Karolina Kósa

**Affiliations:** 1Department of Behavioural Sciences, Faculty of Medicine, University of Debrecen, 4032 Debrecen, Hungary; radi.bence.mark@med.unideb.hu (B.M.R.); kosa.karolina@med.unideb.hu (K.K.); 2Doctoral School of Health Sciences, University of Debrecen, 4032 Debrecen, Hungary

**Keywords:** mental health, medical students, help-seeking, online counselling

## Abstract

Medical students are at increased risk for psychological morbidity but the majority of those with mental health problems do not seek professional care. We aimed to uncover the viewpoints of medical students regarding barriers and facilitators to using university mental health services and their attitudes and preferences towards online counselling. Four semi-structured focus groups were conducted (*n* = 26, mean age = 21.8, ±1.88, 73% males). After reaching data saturation, interviews were audio-recorded, transcribed and content-analysed by two independent coders. Intrapersonal barriers emerged to be perceived low risk, excessive self-reliance, lack of belief in the effectiveness of service, lack of openness. Interpersonal factors were the following: assumed long waiting list, insufficient provision of service information, fear of exposure, and not being familiar with the counsellor and the process. Extrapersonal barriers such as insurance problems, the number of available sessions, adverse sociocultural attitudes, fear of stigmatisation were identified. Students suggested that the university should provide psychoeducation and routine screening, apply social marketing and stigma reduction campaigns, improve information flow, and offer not only personal but also online video counselling to target removing these barriers. The results provide a reference for the redesign of mental health services to facilitate their access by students. Implications and limitations are discussed.

## 1. Introduction

The years spent in higher education are of utmost importance for students who finalise their transition from childhood to adult life during this time. Taking up the responsibilities of adulthood combined with the burdens and expectations related to studies creates stressful life experiences. Medical students are especially burdened due to high workloads, individual responsibilities, and related stress associated with their studies [1]. The mental status of medical students has generated increasing interest since the early 1980s when Lloyd and Gartrell [2] found considerably higher prevalence of psychiatric symptoms in US medical students compared to those in the general population. In addition, Firth revealed [3] that only 24% of British medical students were free of psychiatric symptoms as opposed to 68% of the control consisting of young, employed people.

Increasing interest is reflected by the fact that only five publications were found between 1981 and 2000 having the major topics of “medical students” and “mental health” whereas 180 papers were identified having the same search terms between 2001 and 2020 (PubMed search, Accessed 1 March 2022).

As of 2022, a great volume of research supports the conclusion that mental health problems including depression and anxiety have been prevalent among medical students globally, including in North America [4], Europe [5], and Asia [6], and even more so than among other university students in non-medical programmes. The proportion of affected medical students is considerable: the overall estimated prevalence of depression or depressive symptoms was 27.2% in a recent systematic review of 195 studies from 43 countries in a sample of more than 120,000 students [7].

College students, in general, consistently demonstrate low levels of help-seeking behaviour even though mental health services are available at schools. In the World Health Organization World Mental Health Surveys, 83.6% of college students with 12-month mental disorders did not receive any 12-month healthcare treatment for their condition [8]. Treatment, if accessed, is typically initiated years after the onset of mental problems [9] although prolonged duration of untreated mental disorders can lead to poorer quality of life and poorer clinical outcomes [10]. There are significantly fewer studies providing information specifically on medical students’ use of mental health services. However, based on the available evidence, this population also has a notable unmet need for mental health care [11,12]. In a study of a private US medical school involving 322 medical students (57% of the total student population in the year of the study), 15.2% were classified as depressed (Beck Depression Inventory score ≥ 8), of which 75% did not use either medication or counselling to treat their condition [13].

Although a great number of research studies have been dedicated to the description of mental health problems among medical students, a much lower number of publications has addressed the potential barriers of service uptake. Clinical characteristics such as the presence of mood or anxiety, and stress-related disorders and substance use, are associated with reduced help-seeking among college students [14]. Moreover, cyclothymic and anxious temperament traits were shown to be associated with distress [15], and cyclothymic and irritable traits significantly predicted recreational cannabis use among medical students [16]. This raises the possibility that affective temperament may play a role in medical students’ resistance to seek help. In the general college population, barriers mainly include the lack of information about the availability of mental health support services, embarrassment [17], perceived low risk of having mental problems [18], excessive self-reliance to solve problems [19], perceived stigma related to help-seeking [20], and insufficient ability to recognise symptoms [21]. The extrapolation of these results to medical students is limited because medical students differ from other undergraduates in a number of important ways. First, medical students are more concerned about health-promoting behaviour [22]. Resulting from their training, they are supposed to be more familiar with the healthcare system and more exposed to information on the importance of early treatment utilisation. Furthermore, from the very beginning of their medical studies, they are introduced to the biopsychosocial model of health through a range of behavioural science and psychology courses, which also bring them into significant contact with psychology-trained teachers. Several of their taught subjects cover mental health, psychiatric disorders, and risk assessment of various diseases and pathological mental conditions. In a study of 103 medical students and 395 participants from the general population, medical students showed a more positive attitude towards psychological help-seeking [23]. Despite this, the undertreatment of mental health problems among medical students is still concerning and there is a dearth of research investigating the barriers of help-seeking in this population. In PubMed, only 18 articles were found having the search terms “medical students” (or “medical school”) and “help-seeking” in article titles (PubMed search, Accessed 1 May 2022). Of these 18 studies, only three hold relevant results about the potential barriers of help-seeking other than stigma; however, the subjects are either 20 years old, or represent only a special group or limited number of medical students [24,25,26]. After checking reference lists, three more relevant studies were identified, only one of which was from the past 15 years [12,13,27]. In addition to some of the aforementioned general barriers [12,27], the fear of having a record of a mental diagnosis that would affect their future medical career has emerged as a special deterrent to seeking help as a medical student, especially when the university is involved [25,26].

Most of the presumable barriers to seeking psychological help in this population derive from quantitative studies carried out among college students in general, which may not be appropriate for uncovering medical students’ perspectives regarding perceived barriers and desired interventions to service uptake [28]. Given that the Ottawa Charter, the founding document of health promotion, states that those who are involved and/or stakeholders should be involved in health promotion initiatives at all stages of planning, implementation, and evaluation [29], this principle should also be applied to interventions to remove barriers and facilitate help-seeking among medical students. There is very little qualitative research on this topic. Furthermore, as far as we are aware, the direct perspectives of students on the required interventions to promote help-seeking behaviour have not been taken into consideration in any previous studies. In addition, to the best of our knowledge, no study has investigated the barriers of help-seeking among college students in Central Eastern and Eastern Europe.

The University of Debrecen has had two mental health service providers for more than 20 years. One of these services provides mental health support for all University citizens, whereas the other is run by the medical faculty and is available to medical students. A common factor to both services is that students can request four free counselling sessions by registering via the service’s email address. A dispatcher replies to the request within 24 h and the client usually receives an appointment within one week. The care is provided by psychologists. The COVID-19 pandemic further increased the vulnerability of medical students [30,31] and greatly reduced the availability of mental health services and peer support due to the suspension of personal attendance of classes, services, and clinical rotations [32,33]. Several universities, including ours, provided only face-to-face counselling to students before the pandemic, which had to be transferred to cyberspace due to social distancing. This change in the mode of counselling led to increased scientific attention regarding online forms of mental health support [34]. Due to its advantages, it was expected that the online space would reduce stigmatisation and embarrassment linked to help-seeking in person; to date, however, this has not been equivocal [35]. In addition, there is a dearth of information on medical students’ attitudes and openness to e-mental health services [36].

Our study was based on the theoretical framework of patients’ health care decision making by Llewellyn-Thomas [37]. This holistic approach is organised around the interactive elements of three separate, but interlinked, domains that determine health care decisions encountered by patients. At the centre of the model is an “Intrapersonal Rubik’s cube” having three faces: (1) health care aspects of a decision problem: health status, treatment processes, time periods, and participation in decision making; (2) perspectives of the patients: information, expectation, and preferences; (3) clinical, sociodemographic, and psychological characteristics of the patient. The intrapersonal layer includes what the patient contemplates with respect to the decision. This is surrounded by the second layer, the “interpersonal sphere”. This domain considers the actions and features of others directly involved in the process, such as family members and health care professionals, and the information flow between them. The “extrapersonal sphere” surrounds the previous layer, representing the sociocultural, socio-political context and systemic factors that influence the health care decision making.

We aimed to uncover and understand the barriers to seeking mental help among medical students during the pandemic at the University of Debrecen, along with their suggestions to remove or limit these barriers in a way that allowed unlimited freedom of expression. Given the aforementioned characteristics of the academic training and socialisation of medical students, we hypothesised that barriers to help-seeking in medical students were somewhat different from barriers observed in their non-medical student peers. We expected that reasons for not seeking help are rather related to stigmatisation and confidentiality issues in this group, and less related to low levels of risk perception and inability to recognise symptoms. We applied a solution-focused approach to understand not only the barriers, but also the potential starting points for interventions from the students’ perspectives for which we did not find precedents in the literature. Another aim was to assess medical students’ openness and preferences towards online counselling. In this regard, we hypothesised that medical students hold rather positive attitudes towards online counselling due to its more confidential nature.

The ultimate purpose of this study was to provide a comprehensive list of barriers and facilitators of mental help-seeking among medical students, to identify the starting points for interventions in the medical school and thereby promote mental health by reducing the treatment gap.

## 2. Materials and Methods

### 2.1. Study Design and Participants

A qualitative study in the form of focus group interviews was designed to uncover the in-depth perspectives of medical students regarding access and uptake of mental health services at the University of Debrecen. This data collection method gives participants more freedom of response than standardised questionnaire data collection procedures. New aspects and information may emerge that were not initially brought to the attention of the researcher. A wide range of viewpoints can be explored. The method is also time efficient and provides an opportunity to discuss suggestions and build consensus [38]. To obtain information about the medical student population, the interview questions were designed to target what the participants considered to be barriers or facilitators of help-seeking among medical students in general, rather than the factors they were personally hindered by. The research was carried out using the Consolidated criteria for reporting qualitative research (COREQ) guidelines [39]. Four focus groups were planned with no less than six and no more than ten participants per group. Students of general medicine from all six study years and from both courses (one taught in Hungarian and the other taught in English) were invited for voluntary participation using the snowball sampling technique. Focus groups were designed to have a diverse composition in terms of gender and year of attendance, but separate groups were organised for Hungarian- and English-speaking students. Representatives of the Student’s Union received special invitations to the interviews. They play an active role in the community and have a wide social network and insights into the problems of the medical student population. Thus, we found their presence in the sample particularly important to increase the validity of the results.

Ethics approval was issued by the Regional Institutional Research Ethics Committee, Clinical Centre, University of Debrecen under the approval number DE RKEB/IKEB 5821-2021. Informed consent was obtained from all individual participants included in the study. The research was performed in accordance with the Declaration of Helsinki.

### 2.2. Structure of the Interviews

Two major topics were defined for the interviews: I. determinants of stress in medical school and means to address them; and II. access and uptake of existing mental health services, and barriers to use these services. Discussion related to the first and second major topic could be clearly separated in the audio recordings. Results related to the first major topic are under publication elsewhere [40]. Regarding the second topic, the following questions were defined for the semi-structured interview: (1) What are the barriers that may prevent students from using the psychological services available at the University despite difficulties they may be experiencing? (2) For each mentioned barrier, the students were asked how these could be removed or limited. (3) To what extent are medical students open to online counselling? (4) Which online modalities would be preferred by medical students: video or text-based counselling? (5) What is the preference of medical students for online counselling compared to face-to-face counselling?

Semi-structured interviews were conducted according to the pre-designed concept while allowing participants maximum freedom of expression. Subsequent to establishing rapport with the groups, participants were encouraged to interact in order to facilitate discussion and express personal views.

### 2.3. Data Collection

Interviews were conducted during November–December 2021 (the second half of the first semester of 2020/21). Due to the fact that teaching at that time was conducted online due to COVID-19 restrictions, the group interviews also had to be conducted online using videoconferencing software (Cisco Webex 41.4.7.10 for Windows, Cisco Systems, Inc., San Jose, CA, USA), which allowed participants to respond without personal identification. The main topics were communicated to the students when they were recruited and at the beginning of the interview. Prior to the start of the interviews, the students gave their informed voluntary consent to participate and to be audio recorded. No information allowing personal identification was recorded, and students were assured that all information obtained before or during the interviews would be kept anonymous. Sociodemographic data on age, gender, year of attendance, and nationality were collected to ensure diversity.

All interviews were moderated by the first author (MSc in Health Psychology, full-time Ph.D. student) who was helped by the second author (Health Psychology MSc, 2nd yr. student) as assistant moderator. Both belonged to the same age group as the participants (18–25 years) but were in neither informal nor formal relationships with the interviewees or had jurisdiction over them. None of the moderators worked at the University mental health services. For these reasons, it was assumed that students were allowed to express even their negative opinions about the services regardless of the consequences. All participants were encouraged to join the discussion and respect the opinion of other participants. Generally, most (but not all) participants knew at least 1–2 people in the groups. Whenever necessary, the moderator explored a topic in greater depth to steer the conversation towards underlying causes of the perceived barrier(s), and their potential solutions.

### 2.4. Data Analysis

The audio recordings of the group interviews were transcribed verbatim and were transferred to NVivo Content Analysis Software 12 for Windows (QSR International, Burlington, MA, USA) for content analysis performed by two independent coders (the first 2 authors). The approach followed Mayring’s method [41].

Our research questions on the barriers to help-seeking and suggestions to remove them, as specified above, were deductively transformed into main categories (“Barriers”, “Interventions”) and further sub-categories were created forming inductive categories (e.g., “fear of stigmatization”; “more campaigns to reduce stigma”).

Research questions regarding openness towards online counselling and preferences for the different modalities (video, text) were also transformed into 8 main categories: (1.1.) “openness towards online counselling”; (1.2.) “refusal of online counselling”; (2.1.) “preference for video counselling”; (2.2.) “refusal of video counselling”; (2.3.) “preference for text-based counselling”; (2.4.) “refusal of text based counselling”; (3.1.) “preference for online counselling”; (3.2.) “preference for face-to-face counselling”. Then, units of analysis, i.e., any expressions holding a response to the research questions, were grouped into the main categories by deductive coding and the number of units were summarised per category. To ensure replicability, a description of the coding trees of the inductive and deductive category formation (in their order) are presented in Table A1 and Table A2.

The two coders independently coded all four focus groups then compared the two independent category systems and discussed the differences. Few adaptations were made until the final category system was created. The system was supervised by a junior medical doctor to ensure a balanced approach. The senior author reviewed and approved the category system (K.K. is an institute director and professor in preventive medicine and has vast knowledge regarding mental health and the health behaviour of healthcare students, and qualitative research [42,43,44,45]. In the last group, only one new inductive category appeared (“Limited number of counselling sessions”), and responses on openness and preferences for online counselling consistently followed a similar pattern in all four groups. Hence, we considered data saturation reached at this point after four groups (two Hungarian and two English group interviews). Relevant quotes in Hungarian were translated into English after data analysis for quotation in the manuscript. The quotes are identified by the number of the focus group (e.g., FG1), the course (Hungarian or international [English]), and study year of the student (1–6 years).

## 3. Results

The total duration of group interviews was 480 min, of which a total of 120 min was used to discuss the topic of this paper, having a corresponding word count of 15,625 words. Altogether, 26 medical students participated in four focus groups, 13 Hungarian and 13 international students from 10 different countries, from all six years of the course of general medicine: four students from year one, six students from year two, three students from year three, five students from year four, seven students from year five, and one student from year six (internship). Around one-third of the participants (*n* = 8) were representatives of the Student’s Union. Each group consisted of 6–7 participants. The mean age of the participants was 21.8 years, male students comprising the majority (73%). Three students from India, two from Egypt, and one each from Iceland, Jordan, Nigeria, Iraq, Taiwan, Zimbabwe, Vietnam, and Pakistan participated in the research. The sociodemographic characteristics of the interviewees are presented in Table 1.

### 3.1. Types of Barriers to Seeking Mental Help and Suggestions for Interventions

In total, 12 different barriers were identified, which were organised around three domains: intrapersonal, interpersonal, and extrapersonal, in line with the theoretical framework of Llewellyn-Thomas [37]. To limit or remove these barriers, eight different suggestions were identified. Barriers and interventions were paired in the coding system as presented in Table 2.

#### 3.1.1. Intrapersonal Factors

Perceived low risk

Students do not realise the extent and/or severity of their problems and may not be able to judge the point at which they cannot deal with their problems alone and should seek professional help.


*“I’m speaking from my experience, I wouldn’t know when I should go to a psychologist, I don’t know when it’s too much, I don’t know when I’m ill, so to speak.”*
(FG1, HU, Y2)

Students would require help to decide when to turn to professional help.


*“- And it’s also a huge problem that we think that it’s normal, that all of us have the same problem, everybody has the same obstacle, it’s bad for them, it’s bad for me, and I accept that it’s bad for them, so bad is also normal for me.*



*- And we shouldn’t see that as normal, and I think we should tell everyone not to see feeling low as normal, that it’s bad for them.”*
(FG1, HU, Y2 and Y3)


*“They may be interested in some basic guidance on what problems medical students may encounter.”*
(FG1, HU, Y3)


*“You think that you don’t really need it.”*
(FG2, INT, Y1)


*“- (…) just continuing to raise awareness about the importance of mental health.*


*- But raising awareness is not enough, actions are also very important. (…) Don’t just send us emails about mental health, no, we need proper actions, whatever these are I don’t know.”*.(FG2, INT, Y4 and Y1)

2.Excessive self-reliance

Participants expressed the view that medical students have a certain sense of pride of knowing-all that prevents them from turning to someone from another profession for help.


*“Because a medical student really has this pride or even conceit and they think they know everything so they are not going to go to a psychologist, that would be very rare.”*
(FG1, HU, Y2)


*“Even though you might really need help from someone else you say that nah, I can do this by myself, even if you’re at the end of your rope and you might really need [help].”*
(FG2, INT, Y4)

3.Lack of belief in the effectiveness of service

The general attitude towards available services at the University was rather negative. Students expressed the opinion that these services are inadequate and doubted their effectiveness despite having no personal experience.


*“Do you consider the currently available services adequate?”*
(Moderator)


*“I think they are not adequate at all.”*
(FG1, HU, Y2)


*“I don’t think they’re good at all.”*
(FG2, INT, Y4)


*“(…) people are really doubting their effectiveness.”*
(FG4, INT, Y5)

4.Lack of openness

Lack of openness to psychological counselling among students was reported only by Hungarians.


*“I actually think that somehow, they [fellow students] don’t take the first step, they don’t start to care about [their mental health].”*
(FG1, HU, Y2)

Counselling should be made easily available; one means of achieving this would be online counselling. The University moved its counselling service online in April 2020, but participating students were not aware of this. Routine annual screening was also mentioned as an intervention to increase the receptivity of students.


*“[the services] should be easy to access”*
(FG1, HU, Y3)


*“Should be possible to talk over the internet, especially now during COVID-19, such a call would be very helpful for many people, and all you have to do is click on a link.”*
(FG1, HU, Y2)


*“And I*
*think that if it [screening] were mandatory, whether you want it or not, whether you feel it or not, you should go for a routine check-up once every year. I don’t think it would hurt anybody.”*
(FG3, HU, Y5)

#### 3.1.2. Interpersonal Factors

Lack of information about services

Participants generally expressed the view that information about available services was inadequate. Students are not aware of available opportunities of help on campus either due to lack of communication or information available at inappropriate and/or underused channels.


*“If you’re not very involved in community life, you don’t know [that services exist].”*
(FG1, HU, Y2 )

One student at year 4 was of the opinion that the mental health service of the University was only launched last semester although the service has been offered for more than 20 years.


*“-Why do you*
*think students don’t go to the service to ask for psychological consultation? So, what do you think the main barriers are?”*
(Moderator)


*“-The service only started last semester.”*
(FG2, INT, Y4)

Several international students stated that the University did not use adequate communication channels to convey information about its mental health services.


*“They hold a Facebook live [session] where they try to get in touch with students without realizing that a good majority of students are not on Facebook.”*
(FG2, INT, Y6)

In order to improve the uptake of psychological services, it was proposed to promote services extensively online and in person.


*“I think it is essential to keep a very well stacked website that people can really read if they put their minds to it. It should be heavily promoted so that it would pop up in their memory that there is such a thing [service].”*
(FG1, HU, Y1)


*“- What could be a proper action?*
[Moderator]


*- I mean, probably actually coming to the lectures telling us they exist, they started, because it is a well-known fact that no one really reads Neptun [the online information system for students].”*
(FG2, INT, Y6)

2.Assumed long waiting list

International students had concerns about the availability of mental health services of the University. Specifically, they held the view that that there is a waiting list and that it takes a long time to reserve an appointment. This view does not reflect reality, as first appointments are given within one week from the application, or even within 24 h in urgent cases. These concerns also reflect inadequate information flow. Students given appointments who do not show up or cancel their appointments present a problem since they take up time without utilising it.


*“The problem is that they were very limited, and the time slots were limited so he [the student] ended up cancelling that so he was really struggling with it, it was pretty bad.”*
(FG2, INT, Y4)

3.Fear of exposure

In response to the question of what mental health services students should like to be able to access, the first and strongest point among Hungarians was anonymity. This alluded to fear of exposure as a significant barrier in this population. However, fear of exposure did not emerge as an issue among international students. Online access to counselling was mentioned as a potential means to reduce fear of exposure.


*“- What should mental health services be like to get more people to use them?*
[Moderator]


*- Anonymous.*



*- Really, to be able to be completely anonymous, or maybe just talk on the phone, or online, without a camera, etc.”*
(FG1, HU, Y3 and Y2)

4.Lack of familiarity with counsellors/counselling process

Not knowing the counsellors and the counselling process was also mentioned as a barrier to help-seeking for both Hungarian and international students. In spite of the topic covered in their studies, participants did not have clear ideas as to what happens at a psychologist.


*“The other thing is what to expect. It’s okay to go to a psychologist, but what do you expect from them? You don’t know what to expect, and the unknown is something you are afraid of, and that’s why you won’t go.”*
(FG1, HU, Y2)


*“For me it’s also about, like knowing the psychologist as a person, like honestly the psychologist don’t know all the students and students don’t know the psychologists but at least for me I feel it’s hard to open up for someone who doesn’t know me personally.”*
(FG4, INT, Y2)


*“(…) to gain insight so as to make students keener. Or it’s not even about making them keen to go but dare to go [to the first session].”*
(FG3, HU, Y5)

#### 3.1.3. Extrapersonal Factors

Lack of insurance coverage of mental health care

International students must take out health insurance before commencing their studies in medical school; this does not cover mental health care. They can receive a limited number of counselling sessions but, beyond that, they have to pay for private psychological or psychiatric care. Private mental services are available for students for a heavily discounted fee at the University, but students did not seem to be aware of this. They expressed their desire to expand the scope of insurance coverage to include mental health services. (Hungarian students can also receive a limited number of counselling sessions, and their health insurance covers mental health services.)


*“We don’t get any more support unless we have the TAJ card [Hungarian health insurance], which not everyone has … Most of us just can’t afford [private care] because one session with a psychiatrist if needed will cost around 25,000 HUF (appx.70 EUR) without medication.”*
(FG2, INT, Y6)

2.Limited number of counselling sessions

Students can have four counselling sessions for free. Additional sessions are delivered in the form of psychotherapy in accordance with the students’ health insurance. The number of free sessions (four) was considered to be very low by international students, and they expressed a need to increase the number of sessions.


*“(…) I think they should completely remove that limitation.”*
(FG4, INT, Y5)

3.Adverse sociocultural attitudes and fear of stigmatisation

Another set of barriers consisted of the adverse sociocultural attitudes regarding help-seeking in the country of origin and fear stigmatisation. This topic emerged in all groups and its discussion took a significant amount of time. Participants said that students who seek help are negatively perceived not only by peers, but also by themselves (self-stigmatisation).


*“- We don’t have, how to say, the habit of doing this.*



*- And he’s afraid of being looked down upon because he needs help, he’s afraid of asking for help, and he’s afraid of what they’d think of him if they found out that he went to a professional with this problem. And he is already telling himself that if he goes to a psychologist now, he will be less worthy.”*
(FG1, HU, Y2 and Y2)


*“- That’s where I think culture comes in the picture because I mean where most of us come from, I guess mental health is a myth and therapy is like magic. So, for most of us …I think it’s more… we don’t come from that support or believe in that sort of thing. It’s like stigma.*



*- Yeah, I think the stigma is a really big factor because I think… I think for me I mean for me I’m speaking from my personal point of view.”*
(FG4, INT, Y5 and Y4)

Hungarian students recommended routine screening not only as a way to improve mental health, but also as a means to reduce the stigma associated with help-seeking, while English-speaking students highlighted the necessity for reducing stigma and campaigns to raise awareness.


*“- Yes, if for no other reason, it [routine screening of mental health] would be good because maybe there would be more acceptance, because it’s still a big taboo, as you said, so it would be good for that.*



*- If it was a bit compulsory (sic), then people who find it inconvenient or unpleasant but must take it up would be forced into it and wouldn’t be put off by it.”*
(FG3, HU, Y5 and Y5)


*“If we increased awareness among us [international students] to really encourage each other, we can eliminate this [aversion].”*
(FG4, INT, Y5)

### 3.2. Attitudes to Online Counselling

#### 3.2.1. Openness towards Online Counselling

Altogether 13 content-analytical units expressed positive attitudes of medical students to online counselling, whereas only five reflected refusals to participate in such interventions.

Openness towards online counselling: *“Students have now realised its benefits, and many more would be open to it.”*(FG1, HU, Y3)


*“I think online counselling would be nice since they’re in an environment where they feel comfortable (…).”*
(FG2, INT, Y1)

Refusal of online counselling: *“I don’t like doing anything online, I’d prefer to do it in person anyway.”*(FG3, HU, Y5)

#### 3.2.2. Preferences for Different Modes of Online Counselling

Students were of the opinion that text-based counselling may be effective in some cases and its anonymity can increase their help-seeking intention. However, they were consistent in their preference of video counselling compared to text-based counselling. Ten units showed a preference for video counselling, whereas four expressed a dislike for it, of which, three mentioned its non-anonymous nature as the reason for disliking it. Four units expressed preference for text mostly because of anonymity, whereas eight were against it because of its paucity of nonverbal information and assumed lower effectiveness.
Preference for text-based counselling: *“- Another thing is that anonymous letters are also good, because some people, for example, just want to write down their small problems, and they won’t share with their friends or acquaintances because they’re afraid of being looked down upon. On the other hand, if you write it down in a short anonymous letter [to a professional] … Maybe you’re just waiting for them to get back to you, telling you that you’re on the right track, or that you’re not on the right track, what you should do, waiting for feedback, a response.*

*- Or maybe he’s not necessarily waiting for a response…*
*- Just to write it down.*
*- If he gets it off her chest, it’s already easier.”*(FG1, HU, Y2 and Y3)
Preference for video counselling: *“I think it’s better in video than in writing because it matters that we can see the other person’s face and how they say it, because often in writing, even between friends, there are misunderstandings, we don’t know the intonation, we don’t know what they mean so it’s better if we see each other.”*(FG3, HU, Y5)
*“- I think it’s a bit impersonal when someone just typing. (…) So I think chatting might not work for everyone, but I think like a Skype call might be better. I think for some cases.*
*- People would be less opened (to chat instead of a videocall).”*(FG4, INT, Y2 and Y5)


#### 3.2.3. Comparison of Face-to-Face to Online Counselling

Preference for online counselling was expressed in 13 content units, whereas another 13 preferred face-to-face counselling.
Preference for online counselling: *“(…) this [online counselling] is something that many people who would not go to face-to-face therapy would probably be more willing to go to.”* (FG1, HU, Y3)
*“If someone only want to get relief, it is unnecessary to meet in person because it is time-consuming and not cost-effective.”* (FG1, HU, Y2)
Preference for face-to-face counselling: *“The traditional is more intimate, more real.”* (FG1, HU, Y3)
*“Would it be ideal? I think the first option would be better, to have person-to-person type of therapy, but would it be even better to also have an option to go online? Yes, definitely that should be there but I don’t think that one should be the main one they offer, they should have one that is more in person.”* (FG2, INT, Y4)


## 4. Discussion

Our results provided insights into the barriers to help-seeking of medical students at one Hungarian medical school, along with their suggested solutions to remove or reduce these barriers. Among all barriers identified, adverse sociocultural attitudes associated with the fear of stigmatisation and lack of information about available services stand out as major hindrances among both Hungarian and international medical students.

Perceived low risk, excessive self-reliance, and inadequate information about services were also identified in both groups. To the best of our knowledge, our findings on insufficient information about the counsellors and the counselling process as a barrier has not received attention in previous studies. Contrary to previous studies [25,26], the fear of treatment being including in the student’s academic or medical record, and this affecting the future medical career, did not emerge as a barrier in our medical student sample. A possible explanation for this may be that the mental health services of our University are completely independent from the healthcare system and no medical record is created.

Beyond the similarities, differences also emerged between Hungarian and international students. Lack of openness and fear of exposure was more notable among Hungarians. Such extrapersonal barriers as dissatisfaction with the number of available sessions and lack of insurance coverage of mental health care were only expressed by international students. The latter barrier is not easily removed since expansion of the coverage would significantly increase the price of health insurance. These results are in line with data on the use of mental health services at the University, which show that international students are less embarrassed and use these services more frequently than their Hungarian peers. Previous studies conducted in the general college student population showed that perceived stigma [19], insufficient knowledge of mental health services [17], perception of low risk for mental health problems [46], and preference for self-management [18] are widespread. Despite the major differences in their training and socialisation, our results suggest that medical students share mostly common barriers to seeking help at the university mental health services with their non-medical student peers.

To the best of our knowledge, this is the first study directly investigating not only the barriers to seeking help, but also the perspectives of students on interventions to reduce the barriers. Students expressed the need to apply psychoeducational approaches to improve their ability to detect their own mental health risks, to recognise the severity of their mental status, and/or the potential need for treatment, even though studies on mental health, and the risk assessment of pathological mental conditions, are part of the curriculum. Available Internet-and mobile-based risk assessment tools and self-help interventions should be piloted and introduced to help students assess their mental health status and address their potential issues [47]. Routine screening for psychological morbidity may also be contemplated [48], although, in this case, all students who are found to be at increased risk should be referred to appropriate services, which should be able to accommodate all students in need. The first step towards reducing the barriers identified above should be the re-examination of current information channels and platforms used by the mental health service of the University, and/or the creation of one integrated website widely marketed in platforms used by Hungarian and international students alike. Regular provision of up-to-date information for all students not only increases their knowledge, but also helps destigmatise mental health issues and hopefully results in increased student engagement [49]. Making changes to the curriculum and providing earlier contact with mental patients may also reduce stigmatising attitudes.

Medical students were dominantly open to online forms of mental support. Although the possibility of anonymity was seen as a major strength of text-based counselling, its lack seemed to be a deterrent in the case of the video format. However, online video counselling is more likely to be preferred than text-based counselling due to its more informative and personal nature. Nonetheless, individual needs and the nature of the exact problem are also factors that determine the preferred form of counselling, and students’ responses reflected their view that one size does not fit all. Accordingly, modes of mental help should be reflective of the diversity of individuals and their needs. Provision of face-to-face and online counselling are both necessary to respond to needs and effectively reduce the treatment gap.

### Strengths and Limitations

One of the strengths of our research is its qualitative nature that—in contrast to quantitative study methods –provided insight into students’ opinions on sensitive issues. This study design facilitated the understanding of the needs of medical students from their own perspectives, allowing for its wide range. Participants inspired each other and developed ideas together, which was important for producing a list of solutions and potential starting points to promote help-seeking. Uncensored opinions were facilitated by the fact that the moderators were master and doctoral student peers, but had no previous connection with or jurisdiction over the interviewees. Nevertheless, peer desirability and conformity with majority opinion cannot be excluded since some of the interviewees knew each other. Familiarity in the group may also have been a deterrent to raising certain viewpoints. However, this is unlikely in light of our experiences regarding the active participation of group members and the open discussion of some sensitive topics (e.g., one’s own counselling experience as a patient). Potential disadvantages of the online form of communication, such as technical and connection issues, were not prevalent. However, the observation of body language was limited, and participants may have been distracted by external stimuli. Despite having 26 participants involved, data saturation may have been reached. Around one-third of participants were representative members of the Students’ Union, which increased the validity of the results, as these participants are supposed to have an insight into the students’ needs. Conversely, the results are non-quantifiable, and only one university was involved. These factors preclude generalisability; however, since students were asked to share the barriers and facilitators that they considered to be relevant for medical students in general, and not necessarily only the barriers that they personally perceived, the results are appropriate for generating hypotheses about potential points of interventions. There was a dominance of males in the sample. However, since men are well known to be less likely to seek help for mental health problems compared to women, their viewpoints may even be more intriguing [50].

## 5. Conclusions

The findings provide a comprehensive list of barriers to help-seeking at university mental health services among medical students and the starting points for interventions to reduce these. Conclusions emerged directly from the students’ perspectives. To the best of our knowledge, this is the first study conducted on medical students’ barriers to seeking help for mental health that was also aimed at eliciting suggestions to reduce the treatment gap from the students’ perspective. The results show that intrapersonal, interpersonal, and extrapersonal factors mutually play significant roles in students’ help-seeking decision making. The findings also suggest that help-seeking behaviour can be highly promoted by intervening at a systemic level, as most of the barriers emerge from environmental factors, from the interpersonal and extrapersonal domains. We found that, in addition to the well-known fear of stigma and exposure, excessive self-reliance, low risk perception, and difficulties in recognising potential signs of the lack of proper coping mechanisms by medical students, can also play an important role in their avoidance of help-seeking. Lack of belief in the effectiveness of counselling and the need to be familiar with the provider and the process are also barriers to be addressed. The fear of adverse consequences on the future career of psychological help-seeking involving the university is not necessarily a concern for medical students. We did not find other studies investigating the attitudes and preferences of medical students towards online counselling. Our results suggest that medical students may be more receptive to video counselling as a “gate-opener” to obtain a first glimpse into counselling due to higher levels of confidentiality. Offering online counselling, in addition to face-to-face counselling, can contribute to the effective reduction in the treatment gap among medical students and may be kept as an alternative form of service even after the state of emergency due to the pandemic is withdrawn. The findings provide a reference for the redesign and tailoring of the commonly provided mental health services to facilitate their access by medical students.

## Figures and Tables

**Table 1 ijerph-19-07662-t001:** Sociodemographic characteristics of the sample.

Sociodemographic Variables	
Nationality N (%)	
Hungarian	13 (50)
International student	13 (50)
India	3 (12)
Egypt	2 (8)
Iceland, Jordan, Nigeria, Iraq, Taiwan, Zimbabwe, Vietnam, Pakistan	1−1 (4−4)
Age (Mean ± SD)	21.8 (±1.88)
Gender N (%)	
Female	7 (27)
Male	19 (73)
Year of attendance N (%)	
1st	4 (15)
2nd	6 (23)
3rd	3 (12)
4th	5 (19)
5th	7 (27)
6th	1 (4)

**Table 2 ijerph-19-07662-t002:** Barriers of help-seeking at the University mental health services, and suggestions to reduce or eliminate them.

	Barriers	Interventions
Intrapersonal factors	Perceived low risk	Psychoeducation Online and personal promotion of services
	Excessive self-reliance	Psychoeducation Online and personal promotion of services
	Lack of belief in the effectiveness of service	Online and personal promotion of services
	Lack of openness	Online form of counselling Online and personal promotion of services
Interpersonal factors	Lack of information about services	Improving information flow
	Assumed long waiting list	
	Fear of exposure	Online form of counselling
	Lack of familiarity with the process of counselling	Psychoeducation Routine screening
Extrapersonal factors	Lack of insurance coverage of mental health care	Expansion of the scope of insurance coverage
	Limited number of counselling sessions	Increasing the number of counselling sessions
	Adverse sociocultural attitudes	Routine screening
	Fear of stigmatisation	Routine screening Campaigns to reduce stigma

## Data Availability

The anonymised datasets used and/or analysed during the current study are available from the corresponding author on reasonable request.

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
