# Peer review of "Barriers to Seeking Mental Help and Interventions to Remove Them in Medical School during the COVID-19 Pandemic: Perspectives of Students"

_ijerph, 2022, doi:10.3390/ijerph19137662_

Round 1

Reviewer 1 Report

In this manuscript, the author investigated medical student attitude toward using a university mental health service and online counseling. To achieve their goal, they conducted four semi-structured focus groups, the interviews were audio recorded, transcribed, and analyzed by two independent investigators. Results disclosed intrapersonal factors (perceived low risk, excessive self-reliance, and lack of belief of the effectiveness of the mental health services), interpersonal factors (long waiting lists, insufficient information, fear of exposure, lack of familiarity with the counseling process), extra personal factors (number of sessions, stigma, insurance issues) that affect student’ seeking mental health support. The results suggest the need to improve mental health programs making them more accessible by students.

The study is interesting and has strengths and limitations. I have few suggestions for improving the manuscript.

1)      Introductions: the authors should discuss additional factors associated with medical student’s mental health that might affect their attitude toward mental health services. For instances,  affective temperament traits and student’s emotional reactivity affect medical students mental health and might play a role in their resistance of seeking support.  Individual temperament traits are even more important in a qualitative study. Due to the interactive, dynamic nature of group discussions, participants with specific traits may be reluctant to voice dissenting opinions and just want to go along with the prevailing mood.

The authors should discuss this issue and include relevant references:

A)      Infortuna C. Exploring the Gender Difference and Predictors of Perceived Stress among Students Enrolled in Different Medical Programs: A Cross-Sectional Study. Int J Environ Res PublicHealth. 2020 Sep 11;17(18):6647.

2)      Health risk behaviors associated with stress in medical students potentially might increase the stigma and consequent resistance of seeking support. The author should include supporting references:

A)Infortuna C. Affective Temperament Traits and Age- Predicted Recreational Cannabis Use in Medical Students: A Cross-Sectional Study. Int J Environ Res Public Health. 2020 Jul ;17(13):4836.

3)      Where the moderators external or university employees? Were they mental health professionals? For internal qualitative studies, students may give a popular answer that peers and University personnel may agree with rather than a true opinion. This can negatively influence the outcome of the study. In addition, they should discuss potential biased of an online focus group.

4)      The authors should discuss some more the strengths of their research method (compared to a cross-sectional design often used in  other similar studies).The authors should emphasize the novelty of their findings

Author Response

Dear Reviewer,

We would like to thank you for your valuable comments which helped improve the quality of our manuscript. Please see our responses below.

R: Introduction: the authors should discuss additional factors associated with medical student’s mental health that might affect their attitude toward mental health services. For instance, affective temperament traits and student’s emotional reactivity affect medical students mental health and might play a role in their resistance of seeking support.  Individual temperament traits are even more important in a qualitative study. Due to the interactive, dynamic nature of group discussions, participants with specific traits may be reluctant to voice dissenting opinions and just want to go along with the prevailing mood.

  1. The authors should discuss this issue and include relevant references:
  2. A)  Infortuna C. Exploring the Gender Difference and Predictors of Perceived Stress among Students Enrolled in Different Medical Programs: A Cross-Sectional Study. Int J Environ Res PublicHealth. 2020 Sep 11;17(18):6647.
  3. Health risk behaviors associated with stress in medical students potentially might increase the stigma and consequent resistance of seeking support. The author should include supporting references:
  4. A) Infortuna C. Affective Temperament Traits and Age- Predicted Recreational Cannabis Use in Medical Students: A Cross-Sectional Study. Int J Environ Res Public Health. 2020 Jul ;17(13):4836.

Thank you for the recommended studies. We have included these points in the Introduction as follows:

L65-71: „Clinical characteristics such as the presence of mood or anxiety and stress-related disorders and substance use are associated with reduced help-seeking among college students [14]. Moreover, cyclothymic and anxious temperament traits were shown to be associated with distress [15], and cyclothymic and irritable traits significantly predicted recreational cannabis use among medical students [16]. This raises the possibility that affective temperament may play a role in medical students’ resistance to seek help.”

  1. Where the moderators external or university employees? Were they mental health professionals? For internal qualitative studies, students may give a popular answer that peers and University personnel may agree with rather than a true opinion. This can negatively influence the outcome of the study. In addition, they should discuss potential biased of an online focus group.

None of the moderators were employees of the University. The chief moderator was a full-time doctoral student of health psychology, the assistant moderator was a master student of health psychology. Neither had previous connection with or jurisdiction over those interviewed. So we think it is reasonable to suppose that interviewees shared their honest and uncensored opinions. It is included in the text in the Methods, from L219 to L225 and in the Discussion, along with the possible limitations (L592-L599). Regarding the disadvantages of the online form of communication we have extended the limitations with the following:

L599-602: “Potential disadvantages of the online form of communication such as technical and connection issues were not prevalent. However, the observation of body language was limited, and participants could be distracted by external stimuli.”

  1. The authors should discuss some more the strengths of their research method (compared to a cross-sectional design often used in other similar studies). The authors should emphasize the novelty of their findings.

Strengths of the study were expanded:

L587-594: “One of the strengths of our research is its qualitative nature that – in contrast to quantitative study methods –provided insight into students’ opinions on sensitive issues. This study design facilitated the understanding of the needs of medical students from their own perspectives allowing for its wide range. Participants inspired each other and developed ideas together which was important to produce a list of solutions and potential starting points to promote help-seeking. Uncensored opinions were facilitated by the fact that the moderators were master and doctoral student peers but with no previous connection with or jurisdiction over the interviewees.”

The novelty of the findings are emphasized in the conclusions:

L617-619: „To the best of our knowledge, this is the first study on medical students’ barriers to seek help for mental health that also aimed at eliciting suggestions to reduce the treatment gap from the students’ perspective.”

L623-628: „We found that besides the well-known fear of stigma and exposure, excessive self-reliance, low risk perception and difficulties to recognize potential signs of the lack of proper coping mechanisms by medical students can also play an important role in their avoidance of help-seeking. Lack of beliefs in the effectiveness of counselling and the need to be familiar with the provider and the process are also barriers to be addressed.”

L630-631: „We have not find other studies investigating the attitudes and preferences of medical students towards online counselling.”

Thank you for your time and effort!

Reviewer 2 Report

This is an interesting manuscript. A great deal of effort and dedication has been put into extracting information from the situation under study. However, with the design followed and the small number of participants, it is very difficult to generalize results. It would be necessary to increase the number of participants and try to analyze the results more exhaustively using other tools that facilitate the replicability of the results. Thank you for your attention.

Author Response

Dear Reviewer,

We would like to thank you for your time and effort. Please see our response below.

R: This is an interesting manuscript. A great deal of effort and dedication has been put into extracting information from the situation under study. However, with the design followed and the small number of participants, it is very difficult to generalize results. It would be necessary to increase the number of participants and try to analyze the results more exhaustively using other tools that facilitate the replicability of the results. Thank you for your attention.

Focus group interviews need to be saturated for being reliable. Saturation can be reached by as low as 4 focus groups or even with less than 26 persons as has been shown in the literature [1,2]., Saturation was reached in our study, as included in the text. Focus group interviews just as other methods of qualitative research have an overarching aim of generating research hypotheses and uncover details that only narrative (not numerical) data can provide. Qualitative studies by and large do not aim at representativity, but our results are appropriate to give reference for the starting points of interventions. Most of the interventions recommended by medical students have been shown to be cost-effective making their list of suggestions easily implementable. Nevertheless, there are articles in refereed journals with smaller sample size. [3,4]

[1]      Guest G, Namey E, McKenna K. How Many Focus Groups Are Enough? Building an Evidence Base for Nonprobability Sample Sizes: Http://DxDoiOrg/101177/1525822X16639015 2016;29:3–22. https://doi.org/10.1177/1525822X16639015.

[2]      Hennink MM, Kaiser BN, Weber MB. What Influences Saturation? Estimating Sample Sizes in Focus Group Research. Qual Health Res 2019;29:1483. https://doi.org/10.1177/1049732318821692.

[3]      Reinhart A, Malzkorn B, Döing C, Beyer I, Jünger J, Bosse HM. Undergraduate medical education amid COVID-19: a qualitative analysis of enablers and barriers to acquiring competencies in distant learning using focus groups. Medical Education Online 2021;26. https://doi.org/10.1080/10872981.2021.1940765/SUPPL_FILE/ZMEO_A_1940765_SM1885.ZIP.

[4]      Schiekirka S, Reinhardt D, Heim S, Fabry G, Pukrop T, Anders S, et al. Student perceptions of evaluation in undergraduate medical education: A qualitative study from one medical school. BMC Med Educ 2012;12.

The method also has several strengths that are relevant to us, which is why we have chosen this design:

L587-594: One of the strengths of our research is its qualitative nature that – in contrast to quantitative study methods –provided insight into students’ opinions on sensitive issues. This study design facilitated the understanding of the needs of medical students from their own perspectives allowing for its wide range. Participants inspired each other and developed ideas together which was important to produce a list of solutions and potential starting points to promote help-seeking. Uncensored opinions were facilitated by the fact that the moderators were master and doctoral student peers but with no previous connection with or jurisdiction over the interviewees.

Yours Sincerely,

Reviewer 3 Report

Overall, this is a well-written and timely study with important implications. A few minor issues should be addressed:

1. On line 81, a reference is made to a PubMed article search; clarify if this is the same search as noted on line 42 (and if so, why the reported numbers differ), and the time frame for the search.

2. You should be more specific, in terms of your hypothesis (line 138) that medical students would present "somewhat different" barriers than the general population: in what way would they differ?

Author Response

Dear Reviewer,

We would like to thank you for your valuable comments which helped improve the quality of our manuscript. Please see our responses below.

R: Overall, this is a well-written and timely study with important implications. A few minor issues should be addressed:

  1. On line 81, a reference is made to a PubMed article search; clarify if this is the same search as noted on line 42 (and if so, why the reported numbers differ), and the time frame for the search.

Thank you for your recommendation. The reported numbers refer to different papers as the search terms and periods were different. For the 1st search, we used the terms „medical students” and „mental health” to demonstrate the increasing scientific interest in the mental health of medical students in the two decades following the Millenium compared to the period between 1981 and 2000. For the 2nd search, we used the terms „medical students” and „help-seeking” with no time period restrictions to demonstrate the dearth of research investigating the barriers of help-seeking in this population

  1. You should be more specific, in terms of your hypothesis (line 138) that medical students would present "somewhat different" barriers than the general population: in what way would they differ?

The hypothesis and its foundation were further specified in the Introduction:

L148-150: „We expected that reasons for not seeking help are rather related to stigmatization and confidentiality issues in this group, and less related to low levels of risk perception and inability to recognize symptoms.”

L83-84: „Several of their taught subjects cover mental health, psychiatric disorders, and risk assessment of various diseases and pathological mental conditions.”

Yours Sincerely,

Reviewer 4 Report

Dear author

Congratulations to the authors for this pertinent research. Knowing what the barriers to access to health are is essential to develop intervention programs in the future.

I only have a couple of comments

Line 54 reference in APA

What was the degree of agreement of the coders and how are the categories followed?

Kind regard

Author Response

Dear Reviewer,

We would like to thank you for your valuable comments which helped improve the quality of our manuscript. Please see our responses below.

R: Congratulations to the authors for this pertinent research. Knowing what the barriers to access to health are is essential to develop intervention programs in the future. I only have a couple of comments:

1: Line 54 reference in APA

Thank you, the reference was corrected.

2: What was the degree of agreement of the coders and how are the categories followed?

There were only a few, insignificant differences between the coders as follows:

“Lack of belief in the effectiveness of service” was identified by only one of coders. After discussion, the category was included.

“Adverse sociocultural attitudes”: one coder allocated these items in the category of “fear of stigmatization”, but after discussion, they were separated into different categories.

Regarding the preference for providing help online or in personal meetings, one coder identified 14 analytical units to prefer online format, and the other coder identified 13 such analytical units. After discussion, the agreement came to 13 units.

In Appendix the identified categories are shown now in their order of formation. An indication has been added to the text as well: L249-250: “To ensure replicability, a description of the coding trees of the inductive and deductive category formation (in their order) are presented in Table A1 and Table A2.”

Yours Sincerely,

Round 2

Reviewer 2 Report

Thank you for improving the manuscript and justifying the points.